# Efficient Bone Metastasis Diagnosis in Bone Scintigraphy Using a Fast Convolutional Neural Network Architecture

**DOI:** 10.3390/diagnostics10080532

**Published:** 2020-07-30

**Authors:** Nikolaos Papandrianos, Elpiniki Papageorgiou, Athanasios Anagnostis, Konstantinos Papageorgiou

**Affiliations:** 1Former Nursing Department, University of Thessaly, 35100 Lamia, Greece; npapandrianos@uth.gr; 2Department of Energy Systems, Faculty of Technology, University of Thessaly, Geopolis Campus, Larissa-Trikala Ring Road, 41500 Larissa, Greece; 3Center for Research and Technology—Hellas (CERTH), Institute for Bio-Economy and Agri-Technology (iBO), 57001 Thessaloniki, Greece; athananagno@uth.gr; 4Department of Computer Science and Telecommunications, University of Thessaly, 35131 Lamia, Greece; konpapageorgiou@uth.gr

**Keywords:** bone metastasis, prostate cancer, nuclear imaging, bone scintigraphy, deep learning, image classification, convolutional neural networks

## Abstract

(1) Background: Bone metastasis is among diseases that frequently appear in breast, lung and prostate cancer; the most popular imaging method of screening in metastasis is bone scintigraphy and presents very high sensitivity (95%). In the context of image recognition, this work investigates convolutional neural networks (CNNs), which are an efficient type of deep neural networks, to sort out the diagnosis problem of bone metastasis on prostate cancer patients; (2) Methods: As a deep learning model, CNN is able to extract the feature of an image and use this feature to classify images. It is widely applied in medical image classification. This study is devoted to developing a robust CNN model that efficiently and fast classifies bone scintigraphy images of patients suffering from prostate cancer, by determining whether or not they develop metastasis of prostate cancer. The retrospective study included 778 sequential male patients who underwent whole-body bone scans. A nuclear medicine physician classified all the cases into three categories: (a) benign, (b) malignant and (c) degenerative, which were used as gold standard; (3) Results: An efficient and fast CNN architecture was built, based on CNN exploration performance, using whole body scintigraphy images for bone metastasis diagnosis, achieving a high prediction accuracy. The results showed that the method is sufficiently precise when it comes to differentiate a bone metastasis case from other either degenerative changes or normal tissue cases (overall classification accuracy = 91.61% ± 2.46%). The accuracy of prostate patient cases identification regarding normal, malignant and degenerative changes was 91.3%, 94.7% and 88.6%, respectively. To strengthen the outcomes of this study the authors further compared the best performing CNN method to other popular CNN architectures for medical imaging, like ResNet50, VGG16, GoogleNet and MobileNet, as clearly reported in the literature; and (4) Conclusions: The remarkable outcome of this study is the ability of the method for an easier and more precise interpretation of whole-body images, with effects on the diagnosis accuracy and decision making on the treatment to be applied.

## 1. Introduction

Bone metastasis is one of the most frequent cancer complications which mainly emerge in patients suffering from certain types of primary tumors, especially those arising in the breast, prostate and lung [1,2] These types of cancer have great avidity for bone, causing painful and untreatable effects; thus, an early diagnosis is a crucial factor for making treatment decisions and could as well have a significant impact on the progress of the disease, affecting a patient’s quality of life. In the case of prostate cancer diagnosis in men, the metastatic prostate cancer has a significant impact on the quality of their life [3,4]. Lymph nodes and the bones are the sites where metastatic prostate cancer mainly appears. This type of metastasis takes place when cancer cells get away from the tumor in the prostate area and use the lymphatic system or the bloodstream to travel to other areas of a patient’s body [5]. In most men suffering from prostate cancer, bone metastasis mainly sites on the bones of the axial skeleton [6].

Different modern imaging techniques have been used in clinical practice lately, for achieving rapid diagnosis of bone metastases such as scintigraphy, positron emission tomography and whole-body MRI [7,8,9,10]. Even though PET and PET/CT are the most efficient screening techniques for bone metastasis in recent years, due to technological advancements in medical imaging systems, bone scintigraphy (BS) remains the most common imaging method in nuclear medicine and is regarded as the golden standard [11,12,13]. As it is reported in EANM guidelines [14], BS that uses single-photon emission computed tomography (SPECT) imaging, is particularly important for the clinical diagnosis of metastatic cancer, both in men and women [1].

To date, as regards bone scans and other diagnostic images, medical interpretation has been typically conducted by the interpreter who visually assesses the accumulation of a radiopharmaceutical, which has been given to patients. Because this pattern recognition task is based on the experience of the interpreter, computer aided diagnosis (CAD) systems have been introduced to eliminate any possible subjectivity. Their mission is to consider all user-defined criteria for scintigraphy data classification, as well as define a new model for image interpretation purposes.

For the development of CAD systems in medical image analysis domain, machine learning and especially deep learning algorithms have been investigated and implemented, due to their remarkable applicability in diagnostic medical images interpretation [15,16,17,18]. In particular, deep learning has shown exceptional performance in visual object recognition, detection and classification, thus, the development of deep learning tools would be so helpful for nuclear physicians, as these tools could enhance the accuracy of BS [19,20,21].

The contribution of deep learning in medical imaging is attained through the use of convolutional neural networks (CNNs) [16,22,23]. The learning process is an advantageous feature of CNNs as they can learn useful representations of images and other structured data. Until the introduction of CNNs, the task of feature extraction has been accomplished by machine learning models or by hand, whereas medical imaging now uses features learned directly from the data. Judging from the existence of certain features in their structure, CNNs are proven to be powerful deep learning models in the image analysis field [15,16,23,24]. Deriving from its name, a typical CNN comprises one or more filters (i.e., convolutional layers), accompanied by two more layers (an aggregation and pooling layer) that are used for classification purposes [25]. Since a CNN has similar characteristics with a standard artificial neural network (ANN), it uses gradient descent and backpropagation for training tasks, whereas it contains additionally pooling layers along with layers of convolutions. The vector that is sited at the end of the network architecture can deliver the final outputs [17,18,26].

In medical image analysis, the most popular CNNs methods are the following: AlexNet (2012) [27,28], ZFNet (2013) [29], VGGNet16 (2014) [30,31], GoogleNet [32], ResNet (2015) [33], DenseNet (2017) [34]. The advantageous features of CNNs have a significant impact on their performance, making them so far, the most efficient methods for medical image analysis and classification.

Nowadays, the main challenge in BS, as one of the most sensitive methods for imaging in nuclear medicine, is to build an algorithm that automatically identifies whether a patient is suffering from bone metastasis or not, just by looking at whole body scans. The algorithm has to be extremely accurate because the lives of people are at stake. In this direction, CAD systems have been proposed in the domain of nuclear medical imaging analysis, offering promising results. The first CAD system for BS has been introduced in 1997 in [35]. It was semi-automated and allowed quantification of bone metastases from whole body scintigraphy through an image segmentation method, that was proposed in [35]. Fully automated CAD systems were later examined by Sajn et al. [36], who contributed in the diagnostics field by introducing a machine learning-based expert system. Among the proposed CAD systems, there are few already in the market as part of software packages, such as Exini bone and Bonenavi.

Based on the advanced characteristics that deep convolutional networks have, CNNs have been recently applied in nuclear medicine mainly for classification and segmentation tasks showing some promising preliminary results, as presented in [37,38]. Considering bone metastasis diagnosis, CNNs were initially used in BS in nuclear medicine for image analysis and classification using the Exini software [37]. The dataset was delivered by Exini Diagnostics AB and contained image patches of hotpots that had been previously detected. The hotspots in the spine area were the only ones that were deployed to train the CNN, due to lack of time and the ease to classify them. Then, the BSI value was computed after the examined skeleton was fragmented from the background in both the anterior and posterior views. The overall accuracy of the produced results of the examined thesis [37] is assessed as 0.875 for the validation set and 0.89 for the testing set. The second study [38] deals with the exploration of CNNs for classification of prostate cancer metastases using bone scan images, having a significant potential on classifying bone scan images obtained by Exini Diagnostics AB too, including BSI. It entails two tasks, namely: classifying anterior/posterior pose and classifying metastatic/nonmetastatic hotspots. The trained models produce highly accurate results in both tasks and outperformed other methods for all tested body regions when classifying metastatic/nonmetastatic hotspots. The evaluation indicator of the area under receiver operating characteristic (ROC) score was obtained equal to 0.9739, which is significantly higher than the respective ROC of 0.9352, obtained by methods reported in the literature for the same test set.

Focusing on PET and PET–CT imaging [39,40,41], recent studies have shown that a deep learning based system performs as well as nuclear physicians do in standalone mode and improves physicians’ performance in support mode. Although BS is extremely important for the diagnosis of metastatic cancer, there is currently no research study regarding the diagnosis of prostate cancer metastasis from whole body scan images that applies efficient CNNs. A description on previous related works in nuclear medicine imaging, regarding the application of convolutional neural networks, is provided in Section 2.

This study is devoted to the identification of suitable methods for BS image processing (i) to diagnose bone metastasis disease by investigating efficient CNN based methods and (ii) to assess the method performances, based on whole body images. After a thorough CNN investigation regarding the architecture and configuration of hyperparameters, the authors ended up to the proposal of a robust CNN which could identify the right category of bone metastasis. The classification task is a three-class problem that classifies images in healthy, malignant, as well as healthy images with degenerative changes. A thorough comparative analysis between the proposed CNN method based on grayscale mode and a number of popular CNN architectures was conducted to validate the proposed classification methodology. The advanced structures of CNNs, like ResNet50, VGG16, Inception V3, Xception and MobileNet, have been previously proposed for medical image classification [16,42]. In this work, we apply these popular and efficient CNN algorithms in our dataset, making the necessary parameterization and regularization. Experimental results reveal that the proposed method achieves superior performance in BS and outperforms other well-known methods of CNNs. In addition, the proposed method performs as a simple and fast classification method with optimum performance and low computational time.

This study presents innovative ideas and mainly contributes to the following issues:The development of a robust CNN algorithm which can automatically identify metastatic patients of prostate cancer by examining whole body scans;The efficacy of grayscale mode that enhances the classification accuracy compared to other CNN models;A comparative analysis of common image classification CNN architectures, like ResNet50, VGG16, GoogleNet and MobileNet;The establishment of future research directions which will extend the applicability of the method to other types of scintigraphy.

## 2. Materials and Methods

### 2.1. Prostate Cancer Patient Images

All procedures in this study were in accordance with the Declaration of Helsinki. The retrospective review that was performed, comprised 908 patients’ whole-body scintigraphy images from 817 different male patients who visited the Nuclear Medicine Department of the Diagnostic Medical Center “Diagnostiko-Iatriki A.E.”, Larissa, Greece, between June 2013 and June 2018. The selected images came from prostate cancer patients with suspected bone metastatic disease, who had undergone whole-body scintigraphy.

Due to the fact that whole body scan images contain some artifacts and non-related to bone uptake, such as urine contamination and medical accessories (i.e., urinary catheters) [43], as well as the frequent visible site of radiopharmaceutical injection [25], it is necessary to remove these artifacts and non-osseous uptake from the original images by following a preprocessing approach. This method was applied by a nuclear medicine physician before the use of dataset in the proposed classification approach.

The initial dataset of 908 images contained not only bone metastasis present and absent patient’s cases suffering from prostate cancer, but also degenerative lesions. Due to this reason and for the purposes of this research study to cope with a three-class classification problem, a preselection process concerning images of healthy patients, patients with degenerative lesions and malignant patients was accomplished. In specific, 778 out of 908 consecutive whole-body scintigraphy images of men from 817 different patients were selected and diagnosed accordingly by a nuclear medicine specialist with 15 years’ experience in bone scan interpretation. Out of 778 bone scan images, 328 bone scans concern men patients with bone metastasis, 271 benign cases that include degenerative changes and 179 normal male patients without bone metastasis. A nuclear medicine physician classified all these cases into three categories: (1) normal (metastasis absent), (2) degenerative (no metastasis, but image includes degenerative lesions/changes) and (3) malignant (metastasis present), which were used as golden standard (see Figure 1). The metastatic images were confirmed after further examinations performed by CT/MRI.

A Siemens gamma camera Symbia S series SPECT System (by a dedicated workstation and software Syngo VE32B) with two heads with low energy high-resolution (LEHR) collimators was used for patient scanning. In total, 778 planar bone scan images from patients with known P–Ca were retrospectively reviewed. A whole-body field was used to record anterior and posterior views digitally with a resolution of 1024 × 256 pixels.

### 2.2. Methodology

In this research study, an approach based on CNN model, as a common method in medical image analysis, is suggested for a three-class bone metastasis classification. The three-class classification problem regards bone metastasis presence, absence and cases that are linked to degenerative changes in 778 samples of male patients suffering from prostate cancer and consists of three phases, namely: preprocessing, network design and testing/evaluation. Figure 2 demonstrates the whole process for the examined dataset of 778 patients classifying them into three categories, (i) malignant (with bone metastasis presence), (ii) healthy (no bone metastasis) and (iii) degenerative (healthy, presenting degenerative changes).

#### 2.2.1. Preprocessing Stage

In this stage, images given by a nuclear medicine doctor are first loaded in RGB mode and stored in a PC memory. They are in RGB mode by default and can be transformed to grayscale mode only if requested. Depending on their class, they are given the prefix 1 (one), 2 (two) or 3 (three) as an output, determining whether they are malignant, degenerative or healthy, respectively. Then, data normalization follows, which is a useful and necessary method in a machine–learning process that rescales the data values within 0 and 1. This technique makes the dataset scalable and discards possible outliers that usually confuse the algorithm. In order to avoid choosing wrong samples for training and testing, the shuffling method comes to give a random order to data. In the case of small number of data, data augmentation takes over and applies the techniques of rescale, rotation range, zoom range and flip in order to create variant images without generating new ones and to avoid overfitting, as well. Data augmentation is used only on the training phase that follows next. Finally, the original dataset is split in three segmented parts: testing (15%) and the rest (85%) is split in 80% for training and 20% for validation. The training process gives to the model the opportunity to learn patterns, whereas validation is used for normalizing weights. During the testing process, the model is assessed in terms of accuracy and loss.

#### 2.2.2. Network Design Stage

In our effort to determine which is the most proper architecture for the convolutional neural network (CNN), considering image classification, we first need to perform an exploration process, where the most significant parameters need to be defined. The experimentation and trial phase can help the researchers to explore and then choose the most effective network architecture with respect to the number and type of convolutional layers, the number of nodes and pooling layers, filters, the drop rate and the batch size, the number of dense nodes.

Next, a number of functions are defined through bibliographic research. The activation function, which defines the output of the layer and the loss function, which is used for the network’s weights optimization, are the main functions which are involved in the design of the CNN model. The main features of the CNN and its implementation are adequately described in Section 3. The training process refers to the development of a function that describes the desired relation, based on the training data. The validation dataset is next used in the validation process, which aims at error minimization and reveals the prediction efficiency of the proposed method. In addition, it is important for the selected architecture to minimize the training and validation losses in order to stop the validation process and finalize the model.

#### 2.2.3. Evaluation/Testing Stage

The testing data—which were initially split and completely unknown to the model—are used during the testing process. Predictions on each image class are then carried out by the classifier and finally a comparison between the calculated predicted class and the true class is made. In the next step, common performance metrics, such as the testing accuracy, precision, recall, F1-score, sensitivity and specificity of the model are computed to evaluate the classifier [16,22,44]. An error/confusion matrix is also employed to further evaluate the performance of the model and check if it is biased over one or the other class.

### 2.3. Proposed CNN Architecture for Bone Metastasis Classification

In this research study, a CNN architecture is proposed to precisely identify bone metastasis from whole-body scans of men suffering from prostate cancer. The developed CNN will prove its capability to provide high accuracy with a simple and fast architecture for whole-body image classification. Through an extensive CNN exploration process, we conducted experiments with different values for our parameters, like pixels, epochs, drop rate, batch size, number of nodes and layers [26]. In common classic feature extraction techniques, a manual feature selection was required to extract and utilize the appropriate feature. CNNs, resembling in type ANNs, can perform feature extraction techniques automatically by applying multiple filters on the input images and then, through an advanced learning process, they select those that are the most proper for images classification.

In this context, a deep-layer network with 4 convolutional-pooling layers, 2 dense layers followed by a dropout layer, as well as a final output layer with three nodes, is built (see Figure 3).

The images enter the network at various pixels dimensions, starting from 200 × 200 pixels to 400 × 400 pixels. Following the structure of the CNN, the first (input) convolutional layer consists of 3 × 3 filters (kernels) of size 3 × 3, always followed by a max-pooling layer of size 2 × 2 and a dropout layer with 0.2 as dropout rate. The first convolutional layer has 16 filters and each next convolutional layer doubles the numbers of filters, just like the following max-pooling layers. Next, a flattening operation transforms the 2-dimensional matrices to 1-dimensional arrays, in order to run through the hidden fully connected (dense) layer with 32 nodes. To avoid overfitting, a dropout layer was suggested to drop randomly 20% of the learned weights. The final layer is a three-node layer (output layer).

In the suggested CNN architecture, the ReLU function is used in all convolutional and fully connected (dense) layers, whereas the categorical cross-entropy function is the final activation function used in output nodes. Moreover, two performance metrics, concerning accuracy and loss, are used. As regards loss, the categorical cross-entropy function is calculated with an ADAM optimizer, which is an adaptive learning rate optimization algorithm [45]. For model training, the ImageDataGenerator class from Keras was used, offering augmentations on images like rotations, shifting, zoom, flips and more.

### 2.4. Availability of Data

Data are available from the nuclear medicine doctor N.P. upon reasonable request.

## 3. Results

This section reports the findings of this research study based upon the results produced of the methodology applied. For the classification accuracy calculations, each process was repeated 10 times.

### 3.1. Three-Class Classification Problem Using CNN in Grayscale Mode

Certain hardware and software environments were used to implement the experiments, according to the aim of this work. The experiments were performed in a collaborative environment, called Google Colab [46], which is a free Jupyter notebook environment in the cloud. The main reason for selecting the cloud environment of Google Colab is that it supports free GPU acceleration. Additionally, the frameworks Keras 2.0.2 and TensorFlow 2.0.0. were used.

The software that was used in this study also included OpenCV, which was used for loading and manipulating images, Glob, for reading filenames from a folder, Matplotlib, for plot visualizations and finally, Numpy, for all mathematical and array operations. Moreover, Python was used for coding, Keras (with Tensorflow [47]) for programming the CNN whereas, data normalization, data splitting, confusion matrices and classification reports were carried out with Sci-Kit Learn. The computations ranged between 6′ to 8′ per training (epoch) for RGB images (256 × 256 × 3) and 3′ to 4′ per training (epoch) for grayscale images depending on the input.

Following the steps described in Section 2.2, the images are initially loaded in grayscale or RGB mode. Then, normalization, shuffling and data augmentation take place while the training phase follows next. Dataset is next split into training, validation and testing samples.

A meticulous CNN exploration process was accomplished, in which experiments with various convolutional layers, drop rates, epochs, number of dense nodes, pixel sizes and batch sizes were conducted. Different values for image pixel sizes were also examined, such as 200 × 200 × 3, 256 × 256 × 3, 300 × 300 × 3, 350 × 350 × 3, 400 × 400 × 3, as well as various values for batch sizes, such as 8, 16, 32 and 64 were investigated. In addition, variant drop rate values were studied, for example 0.2, 0.4, 0.7 and 0.9, as well as a divergent number of dense nodes, like 16, 32, 64, 128, 256 and 512 was explored. The number of epochs that was tested, ranged from 200 up to 700. The number of convolutional and pooling layers was also investigated, whereas early stopping criterion was considered without producing promising classification accuracies.

It should be pointed out that authors performed many experiments with several convolutional layers, epochs, pixel sizes, drop rates, dense nodes and batch sizes to find the optimum values of all parameters. After a thorough CNN exploration analysis, the best CNN configuration was the following: a CNN with 4 convolutional layers, starting with 16 filters for the first layer, and for each convolutional layer that comes next, the number of filters is doubled (16 -> 32 -> 64 -> 128). The same philosophy is followed for the max-pooling layers that come next. All filters have the dimensions of 3 × 3. This setup is shown in Figure 3, including the best performance parameters.

Table 1 gathers the performance analysis and results for the CNN models with the following specifications: 4 convolutional layers (16, 32, 64, 128), epochs = 300, dropout = 0.2, pixel size = 400 × 400 × 3, dense nodes = 32–16 and different batch sizes (in Google Colab [46]) for 10 runs. Table 2 presents the 5 runs of the best CNN architecture, entailing 4 convolutional layers (16, 32, 64, 128), epochs = 300, dropout = 0.2, pixel size = 400 × 400 × 3, batch size = 32 examining various dense nodes. It is obvious that the dense 32–16 provides the highest classification accuracies.

Following the exploration analysis, the performance analysis and results produced in Google Colab [46], for the suggested CNN model of 4 convolutional layers (16, 32, 64, 128), epochs = 300, pixel size = 400 × 4003, dense nodes = 32–16 for different dropouts, showed that the CNN is not able to perform properly for dropouts > 0.4. Figure 4 presents the average accuracies for the proposed grayscale CNN architecture, containing various pixel sizes. Figure 5 represents the (a) accuracy and (b) loss precision curves for the best grayscale CNN model.

### 3.2. Three-Class Classification Problem Using CNN in RGB Mode

In the current section, the same CNN exploration process for RGB mode as in grayscale mode, was followed. The network parameters that were considered as the best, were: 4 convolutional layers (16–32–64–128), dense = 256–128, batch size = 16, drop-rate = 0.2, pixel size 256 × 256 × 3 and epochs = 500 (see Table 3).

A series of indicative results from the conducted exploration analysis for best network performance regarding the problem of bone metastasis classification, when RGB mode images are used, is gathered in Table 4. Figure 6 illustrates the precision curves for the best CNN in RGB mode.

### 3.3. Comparison with State of the Art CNNs

To further investigate the performance of the proposed CNN architecture, an extensive comparative analysis between state of the art CNNs and our best model, was performed. Well-known CNN architectures, such as (1) ResNet50 [33], (2) VGG16 [30,31], (3) Inception V3 [48], (4) Xception [49] and (5) MobileNet [50] were used and are described as follows:

(1) ResNet50 is a 50-weight layer deep version of ResNet (residual neural network), with 152 layers based on “network-in-network” micro-architectures [33]. ResNet50 has less parameters than the VGG network, demonstrating that extremely deep networks can be trained using standard SGD (and a reasonable initialization function) through the use of residual modules.

(2) VGG16 [31] is an extended version of visual geometry group, (VGG) as it contains 16 weight layers within the architecture. VGGs are usually constructed by using 3 × 3 convolutional layers, which are stacked on top of each other. For the examined problem, the vanilla implementation of VGG16 was not able to produce a satisfactory accuracy (<70%), thus authors investigated a slight fine-tuning on the number of its trainable layers. Thus, after an exploration process, it emerged that by retraining the weights of the last five layers, the model significantly improved its performance, reaching an acceptable classification accuracy up to 89%.

(3) Inception V3 [48], which is Inception’s third installment, includes new factorization ideas. It is a 48-layers deep network which incorporates RMSProp optimizer and computes 1 × 1, 3 × 3 and 5 × 5 convolutions within the same module of the network. Its original architecture is GoogleNet. Szegedy et al. proposed an updated version of the architecture of Inception V3, which was included in Keras’ inception module [48]. This updated version is capable to further boost the classification accuracy of ImageNet.

(4) Xception, which is an extension of the Inception’s architecture, replaces the standard Inception’s modules with depth-wise separable convolutions [49].

(5) MobileNet convolutes each channel separately instead of combining and flattening them all, with the use of depth-wise separable convolutions [50]. Its architecture combines convolutional layers, depth-wise and point-wise layers to a total number of 30.

After performing an extensive exploration of all the provided architectures of popular CNNs, authors defined the respective parameters as follows:Best gray CNN: pixel size (400 × 400 × 3), batch size = 32, dropout = 0.2, 16–32–64–128 dense nodes 32, 16, epochs = 300 (average run time = 938 s);Best RGB CNN: pixel size (256 × 256 × 3), batch size = 16, dropout = 0.2, conv 16–32–64–128 dense nodes 256, 128, epochs = 500 (average run time = 3348 s);VGG16: pixel size (250 × 250 × 3), batch size = 32, dropout = 0.2, flatten, dense nodes 512 × 512, epochs = 200 (average run time = 2786 s);ResNet50: pixel size (250 × 250 × 3), batch size = 64, dropout = 0.2, Global Average pooling, dense nodes 1024 × 1024, epochs = 200 (average run time = 1853 s);MobileNet: pixel size (300 × 300 × 3), batch size = 32, dropout = 0.5, GlobalAveragepooling2D, epochs = 200 (Average Run Tim e = 2361 s);Inception V3: pixel size (250 × 250 × 3), batch size = 32, dropout = 0.7, GlobalAveragepooling2D dense nodes = 1500 × 1500, epochs = 200 (average run time = 2634 s);Xception: pixel size (300 × 300 × 3), batch size = 16, dropout = 0.2, flatten, dense nodes = 512 × 512, epochs = 200 (average run time = 3083 s).

Figure 7a illustrates the classification accuracy for validation and testing for all the examined CNNs, whereas Figure 7b depicts the corresponding loss for validation and testing for all the examined CNNs.

In what follows, Table 5 gathers the results of state-of-the-art CNN models, concerning the malignant disease class performance which are straightforward compared with our best performed CNN configuration, proposed in this research work.

It is important to highlight that authors have conducted an exploratory analysis for all benchmark CNNs for different dropouts, while they have paid special attention to avoid overfitting [26] by reducing the number of weights (in order to find a network complexity, appropriate for the problem). Thus, not only did they select the best performing model for each CNN architecture in terms of accuracy, but at the same time, the best CNN that avoids overfitting in 10 runs.

## 4. Discussion of Results

The problem of diagnosis of bone metastasis in prostate cancer patients has been tackled with the use of CNN algorithms, which are considered applicable and powerful methods for detecting complex visual patterns in the field of medical image analysis. In the current work, a total of 778 images was acquired, that includes almost the same numbers of healthy, degenerative and malignant cases from metastatic prostate cancer patients. Several CNN architectures were tested, leading to the one that performed optimum under all hyperparameter selection cases and regularization methods, emerging classification accuracies ranging from 89.15% up to 94.07%. Authors selected the grayscale CNN model with four convolutional layers, batch size = 32, dropout = 0.2 and 32–16 dense nodes as the simplest, fastest and most optimum performed CNN, with respect to testing accuracy and loss, as well as performing time.

As can be seen in the relevant literature, the ML-based models underfit if the accuracy on the validation set is higher than the accuracy on the training set [26]. The authors of this study have carefully monitored each training process and made sure the model does not underfit (and does not overfit, too). To mitigate the issue of the small dataset, we employed a data augmentation method that performs various pseudorandom transformations on the images of the existing dataset, in order to create more images that can be used for training and validation. The transformations consist of rotations, width and height shifts, flips, zooms and shears on images [26]. We applied as many transformations as possible, without affecting the key characteristics which are used to classify the original image. Normally, underfit models perform poorly in test sets, which is not the case in this study.

After a thorough exploration analysis, grayscale CNN provides equivalent results with the best RGB-CNN counterpart concerning classification accuracy. It is demonstrated in [51] that grayscale images can add value to such studies, despite having similar or lower performance than their RGB counterparts. The reason behind this, is that by lacking color information, the learning process focuses on other types of characteristics that can be generalized better. Thus, grayscale images can build classifiers with lower computation run time, higher accuracy and a more robust behavior, overall.

For a further discussion of the results produced, the classification performance of the proposed model is compared (even indirectly) with that of other CNNs, which were reported in the literature and have been already applied in the particular problem of bone metastasis classification in nuclear medicine. Due to the fact that none of these research studies provide publicly available data, it is not feasible to accomplish a straightforward comparison with them.

Previous works that are highly related to this research study are those carried out in [37] and [38] regarding bone metastasis classification. Both works employ CNNs for classification of metastatic/nonmetastatic hotspots, providing classification accuracies up to 89%. There are more previous works in this domain but are devoted to PET or PET/CT imaging, which is a different imaging modality. According to the related work in bone metastasis diagnosis in prostate cancer patients from whole body scintigraphy images, there is inadequate formation regarding classification accuracy of BS when CNN systems are considered.

To highlight the classification performance of the proposed CNN method, authors performed a straightforward comparison of the accuracy and several other evaluation metrics like precision, recall, sensitivity, specificity and F1 score indicators of our model with those of other state-of-the-art CNNs, commonly used for image classification problems. From the results produced, including those listed in Table 5, which concern the malignant disease class performance, it arises that the proposed algorithm exhibits better or similar performance to other well-known CNN architectures, applied in the specific problem. This study validates the premise that CNNs are algorithms that can offer high accuracy in medical image classification-based problems and can be directly applied on medical imaging where the automatic identification of diseases is crucial for the patients.

The following text outlines the main advantages of the proposed CNN architecture in grayscale mode according to the results thoroughly presented in Section 3. More precisely, gray–CNN:

(1) Can train model efficiently in a small dataset: Regardless the relevant literature, in which CNN can work properly and effectively only when large datasets of medical images are available, in this nuclear medical image analysis problem where the dataset is small, the proposed gray–CNN model is proven that it can be trained efficiently too. Actually, it does not need network pre-training with ImageNet dataset. On the other hand, the well-known CNN methods use the ImageNet dataset which contains 1.4 million images with 1000 classes to perform model pre-training.

(2) Avoids overfitting: A CNN exploratory analysis for the proposed CNN architectures was conducted, that employed or tested to a certain degree, the three most used techniques for regularization, such as reducing the number of weights (to find a network complexity appropriate for the problem), adding constraints to their values (known as weight decay) and modifying dropout (to prevent complex co-adaptations among weights [25]). In this research work, these three methods were accordingly exploited to discover optimally performing networks that generalize well to unseen input examples.

(3) Reduces architecture complexity: gray-CNN that comes with a simpler architecture exhibits better and much faster performance than other well-known CNN architectures in the field of medical image analysis. In addition, both gray and RGB CNN structures operate with less running time, due to their reduced architectural complexity.

The main outcomes of this study can be summarized as follows:The proposed CNN models appear to have significant potential since they exhibit better performance with less running time and simpler architecture than other well-known CNN architectures, considering the case of bone metastasis classification of bone scintigraphy images;The proposed bone-scan CNN performs efficiently, despite the fact that it was trained on a small number of images;Even though Xception emerges similar classification accuracy, Xception performs with higher loss (both in validation and testing) and high computational time.

Some potential limitations of this research study include: (i) the use of a small dataset, since most of the notable accomplishments of deep learning are typically based on very large amounts of data and (ii) no explanation on the decisions is provided. These limitations will be tackled in our future work as highlighted in the last section.

## 5. Conclusions and Future Directions

To sum up, this is the first research study which fully explores the advantageous capabilities of CNNs for bone metastasis diagnosis in prostate cancer patients, using bone scans. This work also proposes a simpler, faster and more accurate set of CNN networks for classification of bone metastasis than those found on the existing literature and other state-of-the-art CNNs in medical image analysis.

Future work is oriented in three different directions: (i) to perform a further investigation of the proposed architecture using more images from prostate cancer patients as well as patients suffering from other types of metastatic cancer, like breast, kidney, lung or thyroid cancer. This will demonstrate the generalization capabilities of the proposed method, (ii) to offer interpretability and transparency in the classification process while simultaneously increase the accuracy of the model. Interpreting predictions becomes more difficult when deep neural networks are applied since they rely on complicated interconnected hierarchical representations of the training data, (iii) to investigate new combined architectures of efficient deep learning and fuzzy cognitive techniques for medical data modeling and image classification in order to enhance explainability in clinical decision making.

## Figures and Tables

**Figure 1 diagnostics-10-00532-f001:**
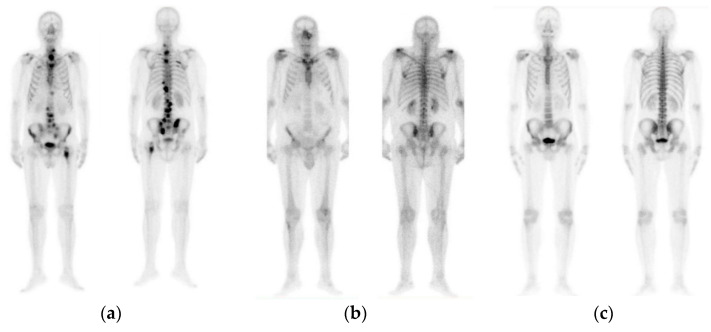
Image samples of prostate cancer in men (from our dataset). (**a**) Metastasis; (**b**) degenerative changes; (**c**) normal.

**Figure 2 diagnostics-10-00532-f002:**
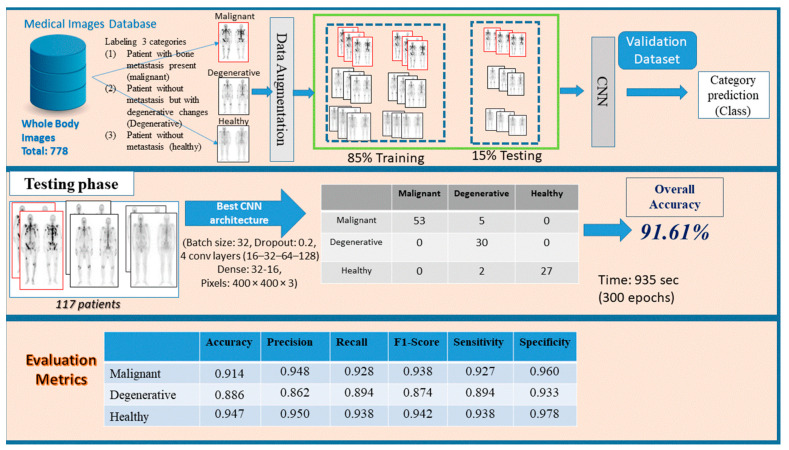
Three stages of the proposed methodology (preprocessing, network design and testing/evaluation) for bone metastasis classification using whole body scans.

**Figure 3 diagnostics-10-00532-f003:**
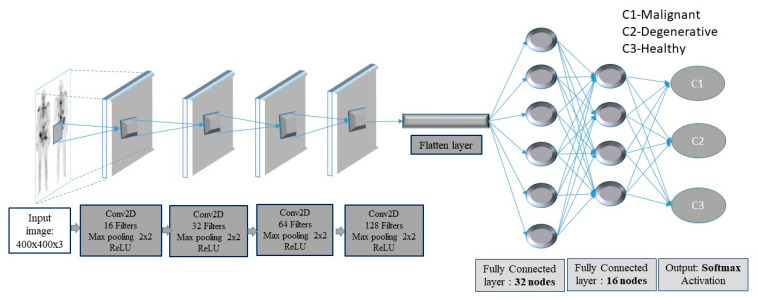
Convolutional neural network (CNN) framework for bone metastasis classification using bone scintigraphy (BS) images.

**Figure 4 diagnostics-10-00532-f004:**
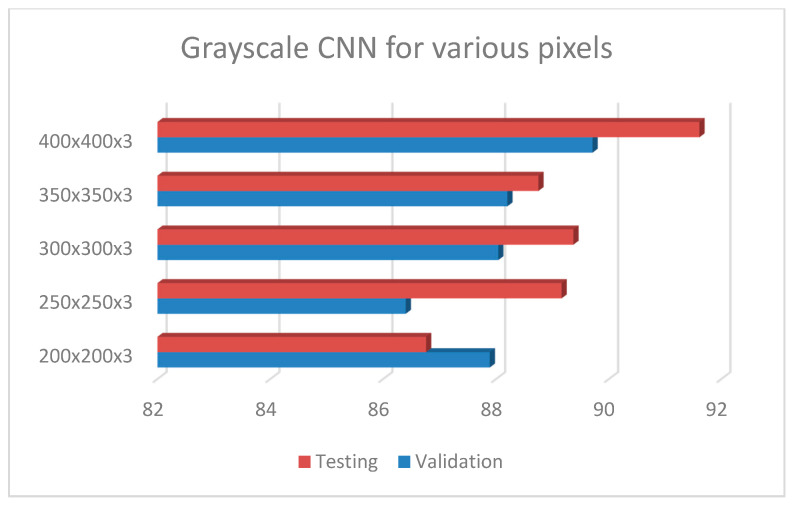
Grayscale CNN architecture with 4 conv (16, 32, 64, 128), epochs = 300, dropout = 0.2, dense = 32–16, dropout = 0.2 and batch size = 32, for different pixel sizes.

**Figure 5 diagnostics-10-00532-f005:**
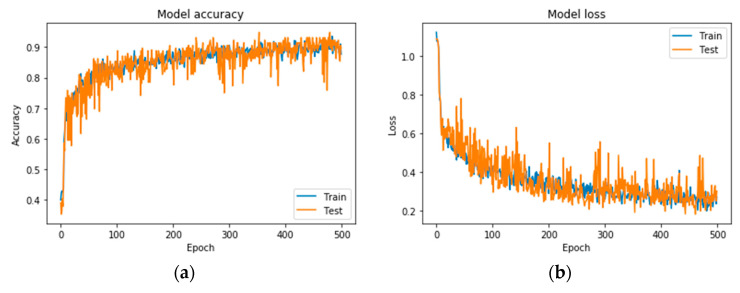
Precision curves for the best CNN in grayscale mode. (**a**) Accuracy and (**b**) loss.

**Figure 6 diagnostics-10-00532-f006:**
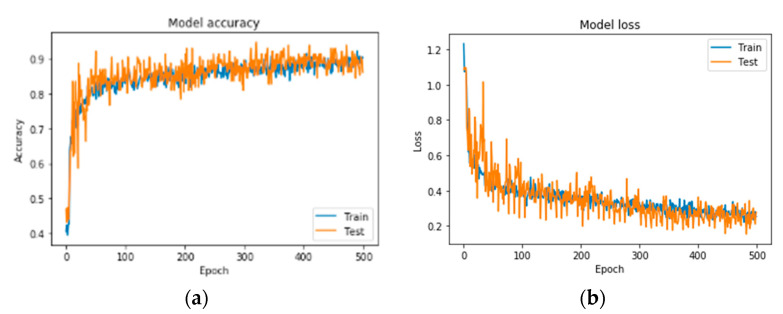
Precision curves for best CNN in RGB mode. (**a**) Accuracy and (**b**) loss.

**Figure 7 diagnostics-10-00532-f007:**
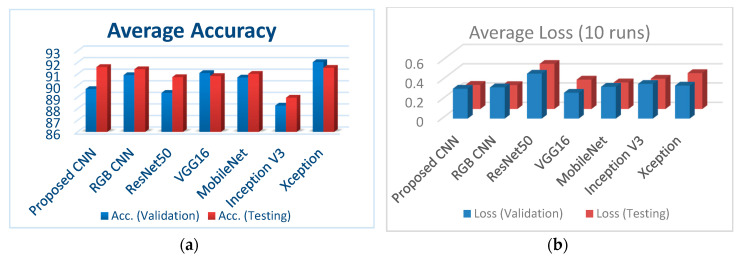
(**a**) Comparison of the classification accuracies for all performed CNNs; (**b**) comparison of loss (validation and testing) for all performed CNNs.

**Table 1 diagnostics-10-00532-t001:** CNN model with 4 conv (16, 32, 64, 128), epochs = 300, dropout = 0.2, pixel size = 400 × 400 × 3, dense nodes= 32–16 and different batch sizes. (AC—accuracy validation; LV—loss validation; AT—accuracy testing; LT—loss testing).

	Batch Size = 8				Batch Size = 16				Batch Size = 32			
	AV	LV	AT	LT	AV	LV	AT	LT	AV	LV	AT	LT
**Run 1**	81.25	0.42	85.72	0.34	92.19	0.3	82.14	0.59	86.16	0.21	89.58	0.13
**Run 2**	92.97	0.17	86.61	0.53	91.41	0.14	92.86	0.26	92.97	0.23	92.71	0.21
**Run 3**	83.59	0.41	95.53	0.19	91.41	1.88	86.61	0.07	89.06	0.25	91.07	0.29
**Run 4**	88.28	0.3	88.39	0.22	77.34	0.38	83.93	0.14	89.84	0.41	96.88	0.09
**Run 5**	82.03	0.34	89.28	0.33	89.84	0.13	88.39	0.29	91.41	0.17	91.67	0.37
**Run 6**	87.5	0.36	89.29	0.27	86.72	0.32	83.04	0.21	89.06	0.10	92.71	0.20
**Run 7**	84.37	0.499	91.07	0.34	86.72	0.39	79.46	0.45	83.59	0.24	90.63	0.19
**Run 8**	90.62	0.24	91.96	0.35	79.69	0.06	89.29	0.43	90.63	0.38	90.63	0.58
**Run 9**	92.19	0.27	88.39	0.4	86.72	0.41	86.61	0.12	90.63	0.97	92.71	0.17
**Run 10**	89.84	0.25	88.39	0.65	89.84	0.3	86.61	0.59	93.75	0.16	87.50	0.24
**Average**	87.26	0.325	89.46	0.36	87.19	0.43	85.89	0.32	89.71	0.31	91.61	0.25

**Table 2 diagnostics-10-00532-t002:** CNN model with 4 conv (16, 32, 64, 128), epochs = 300, dropout = 0.2, pixel size=400 × 400 × 3, dense nodes = 32–16 and different batch sizes; AC—accuracy validation; LV—loss validation; AT—accuracy testing; LT—loss testing.

Dense Nodes	32–16				64–32				128–64				256–128			
	AV	LV	AT	LT	AV	LV	AT	LT	AV	LV	AT	LT	AV	LV	AT	LT
**Run 1**	86.16	0.21	89.58	0.13	85.16	0.21	89.58	0.13	88.28	0.15	86.46	0.48	82.81	0.42	83.33	0.24
**Run 2**	92.97	0.23	92.71	0.21	86.72	0.34	84.38	0.58	82.81	0.30	83.33	0.38	86.72	0.45	89.58	0.37
**Run 3**	89.06	0.25	91.07	0.29	85.94	0.28	89.58	0.30	88.28	0.32	88.54	0.17	89.84	0.12	91.67	0.38
**Run 4**	89.84	0.41	96.88	0.09	92.97	0.15	89.58	0.27	92.97	1.58	86.46	0.38	93.75	0.05	92.71	0.06
**Run 5**	91.41	0.17	91.67	0.37	82.03	0.26	87.50	0.19	89.84	0.50	84.38	0.47	90.63	0.11	84.38	0.34
**AVE**	89.89	0.25	92.38	0.22	86.56	0.24	88.12	0.29	88.44	0.57	85.83	0.38	88.75	0.23	88.33	0.28

**Table 3 diagnostics-10-00532-t003:** CNN architecture for RGB mode (best RGB-CNN).

	Layers	Output Size	Description
	Batch Size: 16, Epochs: 500, Pixel Size 256 × 256 × 3, Dropout: 0.2		
Layer 1	convolution	(none, 254, 254, 16)	filters: 16, kernel size: 3 × 3, input size: 256 × 256 × 3, activation: ReLU
	pooling	(none, 127, 127, 16)	2 × 2 max pooling
	dropout	(none, 127, 127, 16)	drop rate = 0.2
Layer 2	convolution	(none, 125, 125, 32)	filters: 32, kernel size: 3 × 3 activation: ReLU
	pooling	(none, 62, 62, 32)	2 × 2 max pooling
	dropout	(none, 62, 62, 32)	drop rate = 0.2
Layer 3	convolution	(none, 60, 60, 64)	filters: 64, kernel size: 3 × 3, activation: ReLU
	pooling	(none, 30, 30, 64)	2 × 2 max pooling
	dropout	(none, 30, 30, 64)	drop rate = 0.2
Layer 4	convolution	(none, 28, 28, 128)	filters: 128, kernel size: 3 × 3, activation: ReLU
	pooling	(none, 14, 14, 128)	2 × 2 max pooling
	dropout	(none, 14, 14, 128)	drop rate = 0.2
	flatten_1 (flatten)	(none, 25, 088)	
	dense_1 (dense)	(none, 256)	256 nodes
	dropout_18 (dropout)	(none, 256)	drop rate = 0.2
	dense_2 (dense)	(none, 128)	128 nodes
	dense_3 (dense)	(none, 3)	3 nodes, activation Sigmoid

**Table 4 diagnostics-10-00532-t004:** CNN model with 4 conv (16, 32, 64, 128), epochs = 500, dropout = 0.2, pixel size = 256 × 256 × 3, dense nodes = 256–128 and different batch sizes. (AC—accuracy validation; LV—loss validation; AT—accuracy testing; LT—loss testing).

	Batch Size = 8				Batch Size = 16				Batch Size = 32			
	AV	LV	AT	LT	AV	LV	AT	LT	AV	LV	AT	LT
**Run 1**	81.25	0.42	85.72	0.34	91.41	0.23	91.07	0.2	86.72	0.29	92.71	0.18
**Run 2**	92.97	0.17	86.61	0.53	94.53	0.15	91.96	0.25	93.75	0.27	92.71	0.22
**Run 3**	83.59	0.41	95.53	0.19	92.19	0.12	93.75	0.15	91.41	0.27	83.30	0.37
**Run 4**	88.28	0.3	88.39	0.22	91.41	0.22	91.07	0.24	88.28	0.27	92.71	0.21
**Run 5**	82.03	0.34	89.28	0.33	94.53	0.167	90.17	0.253	90.62	0.25	92.71	0.21
**Run 6**	87.5	0.36	89.29	0.27	86.71	0.328	89.28	0.393	87.50	0.44	93.75	0.27
**Run 7**	84.37	0.499	91.07	0.34	90.62	0.22	91.07	0.19	92.97	0.24	92.71	0.19
**Run 8**	90.62	0.24	91.96	0.35	95.31	0.23	93.75	0.139	89.06	0.41	87.50	0.30
**Run 9**	92.19	0.27	88.39	0.4	92.96	0.217	92.85	0.16	85.94	0.37	89.58	0.36
**Run 10**	89.84	0.25	88.39	0.65	85.93	0.45	89.28	0.22	87.50	0.46	82.29	0.39
**Average**	87.26	0.325	89.46	0.36	91.56	0.23	91.42	0.22	89.38	0.33	89.99	0.27

**Table 5 diagnostics-10-00532-t005:** Malignant disease class performance comparison of different methods.

Network	Accuracy (Testing)	Precision	Recall	F1-Score	Sensitivity	Specificity
**Proposed CNN**	91.61	0.948	0.928	0.938	0.927	0.960
**RGB CNN**	91.42	0.935	0.946	0.941	0.946	0.942
**ResNet50**	90.74	0.904	0.972	0.934	0.971	0.921
**VGG16**	90.83	0.952	0.950	0.952	0.949	0.960
**MobileNet**	91.02	0.946	0.941	0.944	0.940	0.952
**Inception V3**	88.96	0.902	0.922	0.911	0.920	0.909
**Xception**	91.54	0.964	0.932	0.946	0.937	0.968

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
