# Peer review of "Efficient Bone Metastasis Diagnosis in Bone Scintigraphy Using a Fast Convolutional Neural Network Architecture"

_diagnostics, 2020, doi:10.3390/diagnostics10080532_

Round 1
Reviewer 1 Report
In the manuscript
„Efficient bone metastasis diagnosis in bone 2 scintigraphy using a fast convolutional neural 3 network architecture”.
the application of convolutional neuronal networks to bone scintigraphy data is investigated. I think this is a very important topic and the presented methodology and the results are of great interest.
While the study is performed very well, the manuscript itself is very long and partly difficult to follow. Therefore, I would suggest to shorten it substantially. Here are my specific comments how to do so:
- Introduction, page 3: The paragraph starting at line 135, “This paper is structured … “ is not necessary at all, please remove.
- Introduction, page 3: The whole section “2. Related Literature in Nuclear Medicine Imaging” is very nice and would be great for a review article, however it is much to long for such a manuscript. Please put the most relevant parts into the point 1 of the introduction and skip the remaining parts.
- Page 7, line 261: It is not important if the study is approved by the director of the Diagnostic Medical Center, so please remove this. Please state instead if there was any approval by a ethics committee?
- Page 8: The Section “2. Main Aspects of Convolutional Neural Networks“, starting at line 301 is a description of Neuronal Networks in general. This is not necessary in the Material and Methods part. Please remove, important aspects can be put into the introduction.
- Page 10: Your section “3. Proposed CNN architecture for bone metastasis classification” should be placed in the Material and Methods section.
- Page 11, Results: There is a very high number of tables and graphs. Please check if this can bed recued.
Author Response
In the manuscript “Efficient bone metastasis diagnosis in bone scintigraphy using a fast convolutional neural network architecture”, the application of convolutional neuronal networks to bone scintigraphy data is investigated. I think this is a very important topic and the presented methodology and the results are of great interest.
We sincerely thank the reviewer for his valuable and insightful comments which were of great help in revising the manuscript. Accordingly, the revised manuscript has been systematically improved, after the following comments were properly addressed.
While the study is performed very well, the manuscript itself is very long and partly difficult to follow. Therefore, I would suggest to shorten it substantially. Here are my specific comments how to do so:
- Introduction, page 3: The paragraph starting at line 135, “This paper is structured … “ is not necessary at all, please remove.
- Response 1: Thank you for the comment. We have deleted this paragraph in the revised manuscript, as not being necessary.
- Introduction, page 3: The whole section “2. Related Literature in Nuclear Medicine Imaging” is very nice and would be great for a review article, however it is much to long for such a manuscript. Please put the most relevant parts into the point 1 of the introduction and skip the remaining parts.
- Response 2: In the revised version of the manuscript, the most relevant parts of section 2 (see manuscript version with tack changes) were included in the Introduction section, whereas the rest of section 2 was deleted accordingly.
- Page 7, line 261: It is not important if the study is approved by the director of the Diagnostic Medical Center, so please remove this. Please state instead if there was any approval by a ethics committee?
- Response 3: Authors have removed this line, after the Reviewer’s suggestion. This study was approved by the Board Committee Director of the Diagnostic Medical Center “Diagnostiko-Iatriki A.E.” Dr. Vasilios Parafestas and the requirement to obtain informed consent was waived by the Director of the Diagnostic Center due to its retrospective nature.
- Page 8: The Section “ Main Aspects of Convolutional Neural Networks“, starting at line 301 is a description of Neuronal Networks in general. This is not necessary in the Material and Methods part. Please remove, important aspects can be put into the introduction.
- Response 4: Section 2 regarding the most important aspects of CNNs, has been integrated in “Introduction”, where it will be better engaged with the content of this section, while the length of the manuscript will be reduced too.
- Page 10: Your section “3. Proposed CNN architecture for bone metastasis classification” should be placed in the Material and Methods section.
- Response 5: We have addressed your concern in the revised version, moving Section 3 to the Material and Methods section.
- Page 11, Results: There is a very high number of tables and graphs. Please check if this can be reduc
- Response 6: Our response to the Reviewer’s comment, regarding the numerous tables and figures, has been as follows:
Figure 2 was deleted as the main phases are also illustrated in Figure 3 (in the initial submitted manuscript).
Table 4 has been deleted, but the outcome of the exploration analysis for different dropouts is mentioned in the text.
Tables 8 and 9 have been deleted, whereas only Table 7 has been kept, as an indicative table for the performance metrics.
Figures 10,11 and 12 have been also deleted, in our effort to reduce the length of the manuscript, without omitting the most significant results produced in this work.
Reviewer 2 Report
Throughout the paper, words are misused in a way that makes it difficult to follow. For example, infected is not a good word to use for metastasis - this refers to an infectious agent. Rather, consider development of metastases, or affected by metastasis.
In addition, the paper is unnecessarily verbose. Much of what is being said can be distilled down to many fewer words.
Overall, it is an interesting paper, just needs some fixing up.
Author Response
Throughout the paper, words are misused in a way that makes it difficult to follow. For example, infected is not a good word to use for metastasis - this refers to an infectious agent. Rather, consider development of metastases, or affected by metastasis.
We thank you very much for your appreciation and positive feedback. Your valuable comments were of great help in revising the manuscript. Accordingly, authors have replaced the word “infected” with that of “developed”, while they have tried to improve the overall content of the manuscript, making it clearer to the reader.
In addition, the paper is unnecessarily verbose. Much of what is being said can be distilled down to many fewer words.
Response 1: Thank you for the comment. The overall length of the revised manuscript has been systematically shortened, according to the Reviewers’ concerns, without omitting the most significant issues of this study. The version with the track-changes will provide a clearer view of the changes that have been made.
Overall, it is an interesting paper, just needs some fixing up.
Response 2: We thank the Reviewer for his insightful comments and positive feedback, approving the manuscript for publication.